# Using a Modified Polysaccharide as a Hemostatic Agent Results in Less Reduction of the Ovarian Reserve after Laparoscopic Surgery of Ovarian Tumors—Prospective Study

**DOI:** 10.3390/medicina59010014

**Published:** 2022-12-21

**Authors:** Rafal Moszynski, Bartosz Burchardt, Stefan Sajdak, Marta Moszynska, Monika Englert-Golon, Piotr Olbromski

**Affiliations:** 1Division of Gynecological Surgery, Department of Gynecology, Obstetrics, and Gynecological Oncology, Poznan University of Medical Sciences, 60-535 Poznan, Poland; 2Department of Forensic Medicine, Poznan University of Medical Sciences, 60-806 Poznan, Poland; 3Student Scientific Society, Poznan University of Medical Sciences, 60-806 Poznan, Poland

**Keywords:** laparoscopy, cystectomy, hemostasis, ovarian function, starch

## Abstract

*Background and Objectives*: The study investigated whether the method of achieving hemostasis affects the ovarian reserve in patients undergoing laparoscopic surgery due to ovarian tumors or cysts. *Materials and Methods*: Patients with unilateral tumors or ovarian cysts, who qualified for laparoscopic tumor enucleation, were randomly selected to receive modified polysaccharides or bipolar coagulation. Ovarian reserve was analyzed by anti-Mullerian hormone (AMH) level. *Results*: The study included 38 patients: 19 patients in the modified polysaccharide group and 19 in the bipolar coagulation group. Patients after bipolar coagulation treatment had statistically significantly lower AMH 6 months after surgery compared to the group treated with modified starch. The levels of AMH in the study and control groups were 3.96 +/− 2.12 vs. 2.51 +/− 1.39 ng/mL, respectively; *p* = 0.018. A statistically significant decrease in AMH was also demonstrated in the bipolar coagulation group as compared to the preoperative assessment (*p* = 0.049). There was no statistically significant decrease in AMH in the group of patients treated with the modified starch. *Conclusions*: Using a modified polysaccharide during laparoscopic cystectomy is effective and has a positive effect on the ovarian reserve compared to the use of bipolar coagulation. Both the AMH level 6 months after surgery and the percentage decrease in AMH were more favorable in the group of patients treated with modified starch.

## 1. Introduction

Ovarian cysts and tumors are significant clinical problems in gynecology. They can consist of functional changes that do not require surgical intervention, they can be non-neoplasm tumors or benign neoplasm tumors, and they can be malignant lesions that require treatment in gynecological oncology centers. The decision to proceed is usually made based on a subjective ultrasound assessment performed by an experienced ultrasound specialist [1]. There is a risk of a false-negative result, but as our previous research has shown, the risk is low (0% for certainly benign and 1.6–2.2% for probably benign) [2]. Endoscopic treatments are widely used if an ovarian tumor is assessed as certainly benign or probably benign. However, it should be remembered that the methods of treatment used should be radical enough to avoid possible recurrence but also maximally conservative so as not to cause excessive damage to the functions of the organ. It is important to eliminate any symptoms of the tumor, prevent its recurrence, and protect fertility and the hormonal function of the ovary. The existence and growth of the tumor itself destroy the healthy part of the ovary. Any operation on the ovary also damages it. Cystectomy itself destroys and, to some extent, reduces the volume of the ovary. Surgical methods are being sought to minimize the damage to ovarian function after the enucleation of the cyst. This is especially important in young women of childbearing age, often before procreation. In cases of benign ovarian tumors, the procedure of choice is the enucleation of the tumor, leaving a healthy part of the ovary. Laparoscopic cystectomy is considered the gold standard in the treatment of benign ovarian tumors [3,4,5]. Such a procedure is particularly justified because it is not known whether an ovarian tumor on the other side will develop in the future. After tumor enucleation, its bed remains, which may, to a greater or lesser extent, be the site of diffuse bleeding. There are many hemostatic methods, the most common of which is diathermy, i.e., the bipolar coagulation of bleeding sites. Another method is to use hemostatic sutures. One of the treatment options for tumor bed bleeding is the use of hemostatic matrix [3] or hemostatic powders [6]. The question of how different hemostatic methods affect the ovarian condition after surgery is interesting, in particular of how the surgical method affects ovarian reserve. In light of contemporary research, it seems that the serum concentration of anti-Mullerian hormone (AMH) best reflects the level of ovarian reserve. It is independent of the cycle phase and endocrine load [7,8,9,10]. AMH is the most accurate. Anti-Mullerian hormone levels drop after surgery but may recover after a few months. A 6-month inspection period is a good choice [7,8,11,12].

### **Aim** 

The study investigated whether the method of achieving hemostasis affects the ovarian reserve in patients undergoing laparoscopic surgery due to ovarian tumors or cysts. It also investigated how other clinical features influence the level of ovarian reserve.

## 2. Materials and Methods

The study included patients with unilateral tumors or ovarian cysts, who qualified for laparoscopic tumor enucleation. The research was conducted at the Gynecological Surgery Department of Poznan University of Medical Sciences, Poland.

Inclusion criteria: age from 20 to 44 years (according to the groups of category standards for AMH used in our laboratory); one-sided tumor of the ovary; qualification for laparoscopic removal of an ovarian tumor; and expressing and signing informed consent to participate in the study.

Exclusion criteria: bleeding requiring additional intervention during surgery, e.g., the need to use a surgical suture; earlier ovarian surgery; hormonal pathologies, e.g., PCOS; use of hormone treatment in the last 3 months; suspected tumor malignancy; past treatment with radiotherapy or chemotherapy; pregnancy; autoimmune diseases; and coagulation disorders.

The research was conducted in a prospective, open-label random selection. Block randomization was used in accordance with the group sizes previously defined in the study plan. Based on the mean in our population group at AMH level as a most important assessment of ovarian reserve and assumed effect size in relation to the control group and power of effect 50% sample size of 19 patients per group was calculated. Randomization consisted of preparing blocks of sealed envelopes in a 1:1 ratio in blocks of 4. A person unrelated to the study and having no knowledge of patients qualified for treatment randomly selected an envelope from the block and provided information on the method of hemostasis to the operator the day before laparoscopy. Patients were qualified either to the study group, in which hemostasis was obtained using a modified starch powder (4DryField^®^ PH), or to the control group, in which classic bipolar electrocoagulation was used. The study was open label because the operator had to know the method to be used. It was possible to coagulate single large vessels in both analyzed groups. If the situation required significantly greater ovarian coagulation or suturing as well as conversion to laparotomy, the patient was excluded from the study.

Assessment of AMH concentration.

Ovarian reserve by anti-Mullerian hormone (AMH) level was analyzed as primary outcome. Venous blood was collected 1–3 days before surgery (AMH 1) and about 6 months after surgery (+/− 7 days) (AMH 2). The blood was centrifuged, and the AMH level was measured in the serum. The concentration of anti-Mullerian hormone (AMH) was determined by the electrochemiluminescence method (ECLIA) using the sandwich immunochemical test, which is a non-competitive method with the use of excess antibodies. In the first stage of the test, the antigen present in the test sample is incubated with biotinylated antibodies specific for the tested antigen and monoclonal antibodies specific for AMH and labeled with a ruthenium complex. Upon addition of streptavidin-coated microparticles, the complex binds to the solid phase through the affinity of biotin and streptavidin. Unbound substances were then removed with a buffer. The voltage applied to the electrode induced an electrochemiluminescence reaction and the emission of 620 nm photons, which were measured with a photomultiplier tube. The concentration of the analyte tested is inversely proportional to the reading of the photon emission.

Analytical Sensitivity–The lower limit of detection for the assay was 0.01 ng/mL.

Surgical procedure.

The therapy was led by the attending physician, and the operations were performed by an experienced surgeon. Laparoscopic surgery was performed by the same gynecologist (RM) under standard conditions (3–4 punctures–5–10 mm, pressure 15 mmHg, Trendelenburg position). After adhesions were released, the cyst was enucleated by traction and contra-traction. It consisted of enucleation with the maximum preservation of the healthy part of the ovary.

For coagulation, bipolar diathermy tweezers from Erbe Elektromedizin, Tubingen, Germany and VIO 300D generators using standard 50 W power were used. Coagulation was applied only as many times and for as long as necessary to achieve hemostasis.

To inhibit bleeding from the cyst bed, a powder of modified starch was also used—a modified polysaccharide of plant origin that has both hemostatic and anti-adhesion effects—4DryField^®^ PH 4DryField PH (PlantTec Medical GmbH, Bad Bevensen, Germany)—in 3 g packages. This substance is in the form of a powder, which, using a dedicated 2-part applicator, is placed on the bed of the enucleated cyst (Figure 1). Hemostasis is quickly and efficiently achieved, and the powder can be slightly moistened to form a gel that has anti-adhesion properties.

Both treatments are registered and used in clinical practice.

Bipolar coagulation is a commonly used method throughout the world. The new method of hemostasis with the use of a polysaccharide has been registered and has been undergoing clinical evaluation for several years, one of the elements of which is our study.

The research was approved by the local Bioethics Committee (No. 266/18).

Tested features:

The following factors were assessed: age of the patient, side affected by the tumor, largest tumor diameter, tumor volume, largest diameter of the contralateral ovary, volume of the contralateral ovary, histopathological diagnosis, and operation time.

The ovarian reserve was analyzed by assessing AMH levels before surgery (AMH 1) and AMH levels 6 months after surgery (AMH 2). According to the literature reports, a 6-month period is optimal for the assessment of possible damage to the ovary, allowing its regeneration in some cases [3].

Statistical methods:

The calculation of sample size for the analysis of the primary endpoint was based on internal Clinical Reports in our setting. A sample size of 19 patients was required to provide 80% power of detecting an effect of 50%.

The calculations were made using Statistica 13.3 (TIBCO Software, Palo Alto, USA) and PQStat v. 1.8.4.136 (PQStat Software, Poznan, Poland). The level of significance was α = 0.05, and the result was considered statistically significant when *p* < α. The normality of the distribution of variables was tested with the Shapiro–Wilk test. Student’s *t*-test, the Cochran–Cox test, or the Mann–Whitney test were used to compare the variables between the 2 groups. To compare the variables between multiple groups, the ANOVA test for related samples or the Kruskal–Wallis test with the Conover–Iman multiple comparison test was calculated. To test whether the changes over the time of AMH were statistically significant, Student’s *t*-test for the related samples or the Wilcoxon test was used.

The relationship between quantitative variables was investigated using the r Pearson linear correlation coefficient or Rs Spearman’s rank correlation. The relationship between the qualitative variables was calculated using Fisher’s exact test or the Fisher–Freeman–Halton test.

## 3. Results

During the study period, between March 2018 and March 2022, 38 patients with ovarian tumors were enrolled and randomly selected. There were 19 patients in the study group and 19 in the control group. The demographic and clinical data are presented in Table 1. The study group and the control group were comparable in terms of these features, and no statistically significant differences were found. It was shown that the groups were comparable and could be subjected to statistical analysis. In the study, 19 women (50%) had lesions on both the right and left sides. There were also no statistically significant differences in this respect between the study group and the control group.

The distribution of histopathological diagnoses in the analyzed women is presented in Table 2.

The age distribution of patients in the groups and the corresponding norms are presented in Table 3.

In the study group, 10 patients (52.6%) showed a decrease in AMH 6 months after the surgery, and in 9 patients the level of AMH 2 was higher. In the control group, 13 patients (68.4%) showed a decrease in AMH 6 months after the surgery, and in 6 patients the level of AMH 2 was higher. Despite the higher percentage of AMH level decrease in the control group, these differences were not statistically significant (*p* = 0.50).

Patients after coagulation treatment had statistically significantly lower AMH 6 months after surgery compared to the group treated with modified starch. The levels of AMH 2 in the study and control groups were 3.96 +/− 2.12 vs. 2.51 +/− 1.39 ng/mL; *p* = 0.018 (Figure 2).

A statistically significant decrease in AMH was also demonstrated in the bipolar coagulation group as compared to the preoperative assessment (*p* = 0.049). There was no statistically significant decrease in AMH in the group of patients treated with the modified starch (Table 4).

It was shown that the absolute decrease in AMH in the control group was greater, but statistical significance was not achieved. However, if the percentage decrease in AMH was taken into account, calculated from the formula (AMH 1 − AMH 2/AMH 1) × 100%, worse results were shown in the group treated with coagulation compared to the hemostatic polysaccharide. In the entire group, the median percentage change in AMH level was a decrease of 6.05%, and the range of these changes ranged from an 84% decrease to a 72% increase. In the study group, the median change meant a decrease in AMH by 2.8% (−84% to + 43%). In the control group, the median change in AMH was a 15.2% decrease (from −84% to + 73%). These differences were significant.

Additionally, statistically significant differences in the baseline AMH levels were demonstrated in individual groups of histopathological diagnoses. Interestingly, these differences were significant between the group of endometrial cysts and the group of teratomas and hemorrhagic cysts. The median AMH1 for endometrioma was statistically significantly higher than in the other groups. These results are presented in Table 5. It should also be emphasized that the group of hemorrhagic cysts was very small.

## 4. Discussion

Surgical treatment consists of removing the diseased organ or its pathologic part. It should be remembered that it should be sufficiently radical, i.e., the diseased tissue should be removed or destroyed in its entirety and sometimes even within the margin of healthy tissues. This guarantees the lowest risk of disease recurrence. Surgical management must also be as conservative and sparing as possible to minimize the loss of organ function and the resulting disability. This also applies to the treatment of ovarian tumors and cysts. There are a lot of papers in the literature on the treatment of ovarian cysts.

Ovarian cyst surgery always causes damage to a larger or smaller part of the healthy parenchyma [3,13].

Moscharini et al. reported on the difference in the impact of different methods such as stripping vs. cystectomy regarding their effect on the ovarian reserve. In their opinion, cystectomy appears to be the most appropriate treatment, both in terms of recurrence and pregnancy rate [14].

Many authors point out that the following factors influence ovarian reserve: tumor size, tumor recurrence, bilateral tumors, applied surgical and hemostatic techniques, and operator skills [4,15,16,17]. Some of these observations, mainly concerning the surgical and hemostatic technique, have been confirmed by our research.

When assessing ovarian function and ovarian reserve, most of the literature concerns endometrioma. It reports that the initial AMH level is lower in patients with endometrial cysts. This was not been confirmed in our studies, where the baseline AMH was significantly higher in endometrial tumors than in other tumors. However, we did not compare this reserve with healthy women without changes in the ovary. Iwase et al. and Chang et al. reported that cystectomy in endometrioma lowers AMH levels more than in other benign tumors [18,19]. Our research results did not confirm this. Importantly, as in the mentioned papers, our research concerned various histopathological types, which is an advantage of the conducted research on the assessment of only endometrial cysts.

In Donnez’s theory, there is a hypothesis that perhaps the coagulation of the endometrial cyst capsule is sufficient instead of tearing it out, in light of the effectiveness of future IVF techniques, but other surgeons report that after enucleation of the capsule there are fewer relapses and still a good response to ovulation stimulation [14,20,21].

There are reports that the enucleation of a cyst is better than the vaporization or coagulation of its capsule, in terms of pregnancy and recurrence prevention, while in terms of ovarian reserve, vaporization is better [4,22,23].

Some authors, such as Donnez, popularize the “sandwich technique—Lap-GnRHa-Lap”. This is a 3-step technique: irrigation in the first laparoscopy followed by gonadotropin-releasing hormone agonists and vaporization in the second laparoscopy, resulting in a lower postsurgical decline of AMH [24].

Ferrero et al. inter alia confirmed that the bipolar coagulation of the tumor bed is simple and widely used, but it destroys the ovary [17]. This is confirmed by the results of our research. Coagulation can also cause other complications, e.g., thermal damage to nearby organs. It also depends on the skill of the operator. Suturing requires more skill [25,26]. Similarly to our studies, Ferrero et al. assessed the percentage decrease in AMH and showed that there were no differences between suturing and coagulation after 3, 6, and 12 months [17]. Takashima et al. obtained similar results on suturing and coagulation [27]. In light of the above, the use of starch seems to be both simple and ovary-sparing, which was confirmed by the results of our research.

Many authors confirm that coagulation techniques destroy healthy ovarian residue [28,29,30]. This was also indirectly confirmed by the results of our research.

The theoretical application of bipolar coagulation is associated with greater lateral destruction of the healthy ovarian tissue and thus greater destruction of its ovarian reserve. There are, however, methods with little lateral thermal destruction, such as the harmonic knife or tools for vessel ligation. However, these methods are less applicable in cystectomy operations, and it is difficult to compare the results, which would be extremely interesting. The harmonic knife finds greater use in the treatment of PCOS [31], and vessel-sealing tools are used more in advanced surgeries such as the removal of the entire ovary or uterus [32].

In a meta-analysis by Ata et al. concerning four studies (213 patients) on endometrial cysts, bipolar coagulation was shown to be more damaging to ovarian reserve as assessed by AMH levels 3 months after surgery compared to ovarian suturing or hemostatic sealant. Coagulation was confirmed to be associated with a greater decrease in AMH 3 months after surgery (by 6.95%) [30].

Cyst enucleation or cystectomy was also reported by Iwase et al. to reduce ovarian reserve [9]. We have shown this to be true in our studies, although in some cases the regeneration of the ovary occurs 6 months after surgery.

In our studies, not all patients had lower AMH levels 6 months after surgery. In some cases, this level was even higher. Sugita et al. stated that the initial decrease in AMH after surgery is regenerated in a certain group of women after 12 months [33]. Similar research results on the AMH level being restored some time after cystectomy were presented by Chang et al. and Lee et al. [18,34]. However, the mean level of AMH decreases, which was confirmed by Ding et al. in their meta-analysis after the elimination of one of the studies. After 6 months, there is a statistically significant decrease in the mean AMH level [35].

In our research, the mean percentage decrease in AMH occurred in all patients, similarly to the study by Asgari et al. [7]. A systematic review by Iwase et al. also reported a general decline in AMH levels and did not provide a clear conclusion as to what is less damaging: coagulation or the suturing of the ovary [9]. Perhaps tight sutures binding the ovary also cause damage through pressure and ischemia.

In the analysis by Ding et al. it was emphasized that suturing the ovary requires greater skill and longer operation time; simultaneously, the use of sutures or hemostatic powders or gels results in fewer adhesions compared to bipolar coagulation [35].

According to the above analysis, the clinical and scientific evaluation of new techniques seems to be significant. Our research is part of this assessment.

There are reports of other hemostatic methods, including one by Song et al., on the comparison of hemostatic sealant with bipolar coagulation, or another by Sonmezer et al. on the hemostatic matrix [36,37]. The results of these studies are to a large extent similar to ours.

There are also reports that compare coagulation with other hemostats or sealants. This mainly applies to TachoSil or FloSeal, i.e., the use of a matrix or gel granules in combination with human thrombin [38]. These studies have shown that all these methods reduce AMH 3 months after surgery, while the decrease is greater with coagulation than with both other methods (41.9% vs. 18.1%, respectively) [38].

Similar results were obtained by Kang et al. (41.2% vs. 16.7%, respectively) for coagulation vs. hemostatic sealants. [39].

These studies assessed the level of AMH 3 months after surgery and concerned slightly different hemostatic methods. Our study assessed the AMH level 6 months after surgery, which, as presented in the introduction, seems to be optimal. Chung’s study, however, showed that one year after surgery with FloSeal application, there were no differences in AMH and FSH concentration compared to the coagulation group [40]. A possible study comparing the use of starch and thrombin-based preparations seems interesting.

Interesting research results were obtained by Kim et al., who showed that the level of AMH 1, 3, and 6 months after the removal of the endometrial cyst by laparotomy was statistically significantly higher than after laparoscopy. They also studied teratoma, for which there were no statistically significant differences [41]. However, it seems that due to other aspects differentiating laparotomy and laparoscopy, such as the formation of adhesions and the number of complications, laparoscopy is the gold standard in the treatment of ovarian cysts.

Torres-de la Roche et al. analyzed the effect on hemostasis and adhesions of the same agent as in our study: 4DryField^®^. The authors showed that this agent had a good effect on lowering the rate of ovarian failure, good hemostasis, and fewer adhesions [6].

Watrowski described the use of 4DryField with a good hemostatic and anti-adhesive effect in three patients treated for gynecological indications [42].

Other studies in the field of gynecology and the use of starch mainly concern the prevention of adhesions, which is an undoubted advantage of 4DryField. Kramer et al. showed that the use of 4DryField in endometrial cyst surgery reduces the formation of adhesions by 85% [43]. On the other hand, studies by Ziegler et al. showed that it is safe from the risk of inflammation and is good at preventing the formation of adhesions despite the increase in C-reactive protein levels after surgery [44].

It is also important to emphasize that there are currently several starch-based hemostatic products on the market (4DryField, Arista, and others). Not all have been studied in humans, and several involve animal model studies. However, comparing their anti-adhesion effectiveness, 4DryField seems to be the most effective [45].

An important element of our research is the evaluation of only unilateral tumors and the randomization of the study, as in the research performed by Asgari et al. [7]. The paper presented by Song et al. relates to bilateral changes and is not based on randomization [28].

The limitation of our research is the relatively small group of patients, especially in different subgroups. However, the inclusion criteria were demanding, and therefore the recruitment time was relatively long. In our research, we also did not assess the length of time and the number of places that were coagulated, which could be the subject of further analysis. It may also be important to compare different hemostatic substances in terms of their effect on the ovarian reserve. It should be noted that this is a pilot study and should be continued on a larger population.

## 5. Conclusions

Using a modified polysaccharide during laparoscopic cystectomy is effective and has a positive effect on the ovarian reserve compared to the use of bipolar coagulation.The AMH level 6 months after surgery is more favorable in the group of patients treated with modified starch. This is especially important in the group of young women who want to maintain the best possible chance of a future pregnancy.

## Figures and Tables

**Figure 1 medicina-59-00014-f001:**
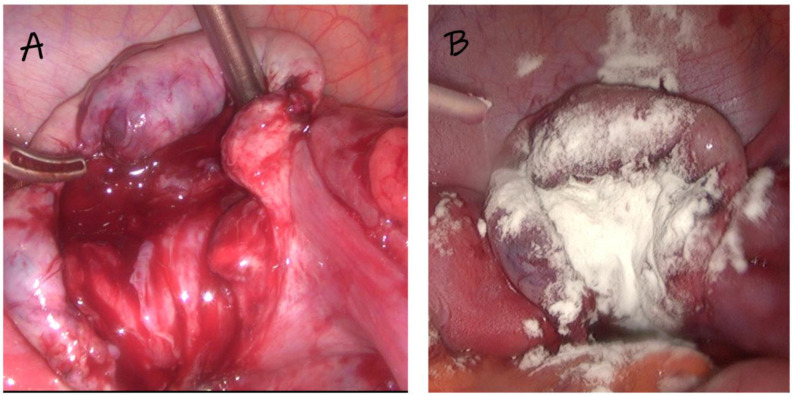
An example of the use of starch on the ovarian tumor bed. (**A**) before use, (**B**) after application.

**Figure 2 medicina-59-00014-f002:**
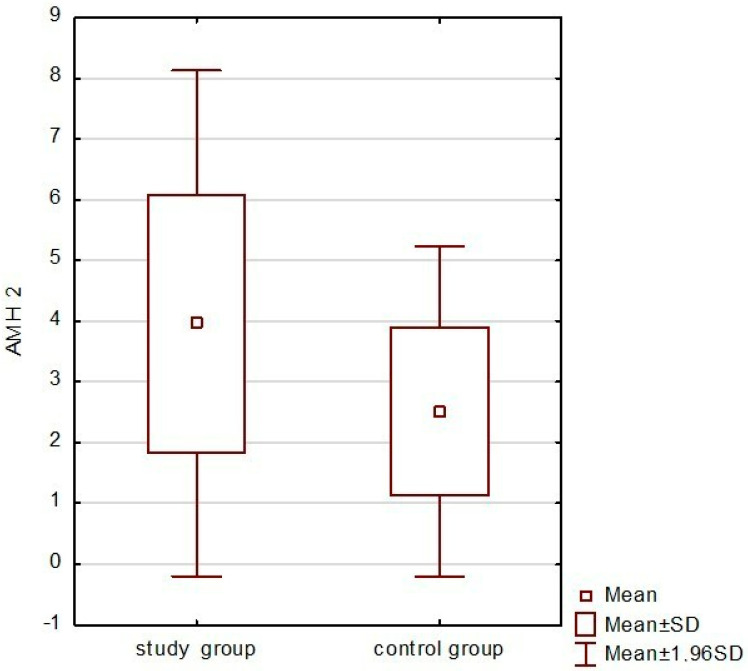
AMH levels 6 months after surgery.

**Table 1 medicina-59-00014-t001:** Demographic and clinical data (mean or median and SD or range).

	Study (Starch) (*N*= 19)	Control (Coagulation) (*N* = 19)	*p* Value
Age (years)	30.8 (+/− 7.05)	30.2 (+/− 7.08)	0.80
AMH 1 (ng/mL)	4.1 (0.78–8.56)	2.9 (0.4–9.78)	0.34
Cyst max. diameter (cm)	4.5 (2.0–12.5)	4.5 (1.9–9.0)	0.98
Cyst volume (cm^3^)	29.0 (3.14–865.14)	44.6 (2.0–220.0)	0.81
Contralateral ovarian max diameter (cm)	2.8 (+/− 0.6)	2.99 (+/− 0.79)	0.49
Contralateral ovarian volume (cm^3^)	8.17 (1.34–20.4)	9.5 (2.8–37.0)	0.90
Duration of the operation (min)	37.0 (25.0–60.0)	40.0 (17.0–82.0)	0.17

**Table 2 medicina-59-00014-t002:** Distribution of histopathological diagnoses in the analyzed groups.

Histopathological Diagnosis	Study (Starch)	Control (Coagulation)	Total
Endometrial cyst	8	9	17
Adult teratoma	6	4	10
Serous cyst	3	5	8
Hemorrhagic cyst	2	1	3
Total	19	19	38

**Table 3 medicina-59-00014-t003:** Distribution of analyzed patients and norms within age groups.

Age Group	All	Study (Starch)	Control (Coagulation)	AMH Norm (ng/mL)
20–24	9	4	5	1.52–9.95
25–29	9	4	5	1.2–9.05
30–34	10	5	5	0.71–7.59
35–39	5	4	1	0.45–6.96
40–44	5	2	3	0.059–4.44
Total	38	19	19	

**Table 4 medicina-59-00014-t004:** Comparison of AMH levels before surgery and 6 months after surgery.

Group		AMH 1	AMH 2	*p* Value
Study (starch)	Mean +/− SD	3.95 +/− 2.25	3.96 +/− 2.12	0.98
Control (coagulation)	Mean +/− SD	3.34 +/− 2.02	2.51 +/− 1.39	**0.049**

**bold** *p*-Value indicates statistical significance.

**Table 5 medicina-59-00014-t005:** Differences in AMH levels before surgery in individual histopathological groups.

Histopathological Diagnosis				*p* Value		
Cyst	Number	AMH 1 Median	AMH 1 Range	Teratoma	Serous	Hemorrhagic
Endometrial	17	4.39	1.89–9.78	**0.005**	0.153	**0.023**
Teratoma	10	2.46	0.79–4.82		0.238	0.665
Serous	8	2.82	0.4–7.48			0.214
Hemorrhagic	3	1.28	1.27–3.59			

**bold** *p*-Value indicates statistical significance.

## Data Availability

Not Applicable.

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
