# Peer review of "Using a Modified Polysaccharide as a Hemostatic Agent Results in Less Reduction of the Ovarian Reserve after Laparoscopic Surgery of Ovarian Tumors—Prospective Study"

_medicina, 2022, doi:10.3390/medicina59010014_

Round 1

Reviewer 1 Report

Interesting paper with a potential clinical impact. Although manuscript has some limitations:

1. Material section require better explanation. Please revise and write how randomisation was made.

2. I propose to join the conclusion that this is a pilot study and should be continued on a larger population.

Author Response

Thank you very much for your important and valuable comments on our research

A more detailed description of the randomization method has been added to the Material section

„This block randomization was used in accordance with the group sizes previously defined in the study plan. Based on mean in our population group in AMH level as a most important assessment of ovarian reserve and assumed effect size in relation to the control group and power of effect 50% sample size of 19 patients per group was calculated. Randomization consisted of preparing blocks of sealed envelopes in a 1: 1 ratio in blocks of 4. A person unrelated to the study and having no knowledge of patients qualified for treatment randomly selected an envelope from the block and provided information on the method of hemostasis to the operator the day before laparoscopy”

We have added the proposed conclusion

“It should be noted that this is a pilot study and should be continued on a larger population.”

Reviewer 2 Report

I think it would be useful to add images from during the laparoscopy and discuss whether the means of coagulation has an influence on the results- to talk a bit about alternative options less damaging to the surrounding cortex could be useful such as harmonic or ligasure.

Author Response

Thank you very much for your important and valuable comments and suggestions regarding our research

We added a laparoscopic image after cyst enucleation and before starch application compared to the image after starch application (Fig1 A , B)

In the discussion, we discussed the theoretical reasons for the greater destruction of the ovary after the use of bipolar coagulation. We also discussed the use of methods such as harmonic knives or vascular ligation methods (e.g. ligasure) in ovarian cyst surgery.

“The theoretical application of bipolar coagulation is associated with greater lateral destruction of healthy ovarian tissue and thus greater destruction of its ovarian reserve. There are, however, methods with a small lateral thermal destruction, such as the harmonic knife or tools for vessel ligation. However, these methods are less applicable in cystectomy operations and it is difficult to compare the results, which would be extremely interesting. The harmonic knife finds greater use in the treatment of PCOS [] and vessel-sealing tools in more advanced surgeries such as removal of the entire ovary or uterus [].”

Reviewer 3 Report

The main finding of this small comparative study is that "patients after bipolar coagulation treatment had statistically significantly lower AMH levels 6 months after surgery compared to the group treated with modified starch".   I identified several major issueas concerning the submitted manuscript:   1) The manuscript would definitely benefit from language editing.  2) What does the study period between "M.2018 and M.2022" mean? What is M.? If this means that this prospective study included only 19 patients per group over four years, a siginficant selection bias is evident. What is the reason for such a small number of included patients? 3) The method descriptions suffer from an extensive description of the laboratory details (probably after inserting the entire manufacturer's instructions into the text) and a scarce, not reproducible description of the surgical procedure. 4) Please provide surgical images that illustrate the manner of application of 4DryField PH and the type (and correctness) of ovarian suture used (e.g. Figure 1a: Ovary before and after hemostatic powder application, Figure 1b: Ovary before and after suturing). 5) The discussion did not reflect any publications addressing the same research question with the same methodology (=AMH levels after ovarian surgery), but using different topical hemostats versus bipolar thermal hemostat. Please discuss the following studies: PMID: 25573183, PMID: 29271052, PMID: 26344349, PMID: 34193356  6) The authors should explicitly embed the experiences with the use of the hemostatic powder in the field of gynecological surgery (PMID: 31480891, PMID: 34036409, PMID: 34357446) 7) The authors should mention that two different modified starch-based agents are available on the market (4DryField and Arista), with 4DryField probably being superior due to its anti-adhesive properties (PMID: 31692813). 8) The limitations of the study should be addressed

Author Response

Thank you very much for your important and valuable comments on our research. We appreciate the insightful and informative support.

Our responses to reviewer comments:

The main finding of this small comparative study is that "patients after bipolar coagulation treatment had statistically significantly lower AMH levels 6 months after surgery compared to the group treated with modified starch".  

Indeed, this is our main significant result, which may be of importance in further research on this topic.

  • „The manuscript would definitely benefit from language editing”

The manuscript was proofread by a paid native speaker once again. But if the reviewer and editor sees a better chance of publishing our manuscript after paid proofreading by a certified company, then of course we will do it.

  • What does the study period between "M.2018 and M.2022" mean? What is M.?

M means March. This has been corrected in the manuscript.

„March 2018 and March 2022”

If this means that this prospective study included only 19 patients per group over four years, a siginficant selection bias is evident. What is the reason for such a small number of included patients?

We are aware of the limitations of our study. We underlined this in the manuscript. Mainly a small study and control group. We planned to conduct this study faster in order to obtain the required number of patients in both groups, but most likely the inclusion and exclusion criteria and the size of the population treated for this reason in our Clinic were too small. We mainly deal with gynecological oncology, in which we also successfully use starch preparations. Also, obtaining informed consent was not automatic. We emphasize in the manuscript that this is a pilot study to encourage other researchers to analyze such a method. We plan to establish inter-center cooperation after this publication.

  • The method descriptions suffer from an extensive description of the laboratory details (probably after inserting the entire manufacturer's instructions into the text) and a scarce, not reproducible description of the surgical procedure.

. We have shortened the description of the laboratory method

The blood was centrifuged, and the AMH level was measured in the serum. The concentration of anti-Mullerian hormone (AMH) was determined by the electrochemiluminescence method (ECLIA) using the sandwich immunochemical test, which is a non-competitive method with the use of excess antibodies. In the first stage of the test, the antigen present in the test sample is incubated with biotinylated antibodies specific for the tested antigen and monoclonal antibodies specific for AMH and labeled with a ruthenium complex. Upon addition of streptavidin-coated microparticles, the complex binds to the solid phase through the affinity of biotin and streptavidin. Unbound substances were then removed with a buffer. The voltage applied to the electrode induced an electrochemiluminescence reaction and the emission of a 620-nm photon, which was measured with a photomultiplier tube. The concentration of the analyte tested is inversely proportional to the reading of the photon emission. Analytical Sensitivity - The lower limit of detection for the assay is 0.01 ng/mL.

The basic conditions of surgical treatment are described. The details and repeatability of course depend on individual clinical situations.

The therapy was led by the attending physician and the operations were performed by an experienced surgeon. Laparoscopic surgery was performed by the same gynecologist (RM) under standard conditions (3-4 punctures - 5-10 mm, pressure 15 mm Hg, Trendelenburg position). After adhesions were released, the cyst was enucleated by traction and contra-traction. It consisted of enucleation with the maximum preservation of a healthy part of the ovary.

For coagulation, bipolar diathermy tweezers from Erbe Elektromedizin, Tubingen, Germany and VIO 300D generators using standard 50 W power were used. Coagulation was applied only as many times and for as long as necessary to achieve hemostasis.”

  • Please provide surgical images that illustrate the manner of application of 4DryField PH and the type (and correctness) of ovarian suture used (e.g. Figure 1a: Ovary before and after hemostatic powder application, Figure 1b: Ovary before and after suturing).

We have added to Figure 1 image A before the starch application and B after the application

In our study, we did not analyze the ovarian reserve after ovary suturing – it was a possible element of exclusion. Therefore, there is no reason to publish images of ovarian suturing and a description of this method.

  • The discussion did not reflect any publications addressing the same research question with the same methodology (=AMH levels after ovarian surgery), but using different topical hemostats versus bipolar thermal hemostat

In our opinion, most of the publications discussed in the discussion concern ovarian reserve (AMH) after various methods of obtaining hemostasis after ovarian cyst enucleation. Most often, this applies to comparisons of coagulation with suturing. There are few comparisons of coagulation with starch in the literature. Therefore, our research seems interesting.

Please discuss the following studies: PMID: 25573183, PMID: 29271052, PMID: 26344349, PMID: 34193356  

We discuss publications proposed by the reviewer:

  1. Publication PMID 25573183 by Ata et al. is quoted. We've added more detailed information:

In a meta-analysis by Ata et al. concerning 4 studies (213 patients) on endometrial cysts, bipolar coagulation was shown to be more damaging to ovarian reserve as assessed by AMH levels 3 months after surgery compared to ovarian suturing or hemostatic sealant. Coagulation was confirmed to be associated with a greater decrease in AMH 3 months after surgery (by 6.95%) [30]

  1. PMID 29271057, PMID 26344349, PMID 34193356

There are also reports that compare coagulation with methods referred to as hemostatic sealants. This mainly applies to TachoSil or FloSil, i.e., the use of a matrix or gel granules in combination with human thrombin [38]. These studies have shown that all these methods reduce AMH 3 months after surgery, while the decrease is greater with coagulation than with both other methods (41.9% vs. 18.1%, respectively) [38].

Similar results were obtained by Kang et al. (41.2% vs. 16.7%, respectively) for coagulation vs. hemostatic sealants. [39].

These studies assessed the level of AMH 3 months after surgery and concerned slightly different hemostatic methods. Our study assessed the AMH level 6 months after surgery, which, as presented in the introduction, seems to be optimal. Chung's study, however, showed that one year after surgery with FloSeal application, there were no differences in AMH and FSH concentration compared to the coagulation group [40]. A possible study comparing the use of starch and thrombin-based preparations seems interesting.

  • The authors should explicitly embed the experiences with the use of the hemostatic powder in the field of gynecological surgery (PMID: 31480891, PMID: 34036409, PMID: 34357446)

In the field of gynecology and the use of 4DryField, we cited the most comparable studies by Torres-de la Roche et al. A new approach to avoid ovarian failure as well function impairing adhesion formation in endometrioma infertility surgery Archives of Gynecology and Obstetrics 2020, 301(5), 1113–1115

Based on the reviewer's suggestion, we discussed other 3 publications on the use of 4DryField in gynecology

  1. PMID 31480891

Watrowski described the use of 4DryField with a good hemostatic and anti-adhesive effect in 3 patients treated for gynecological indications [42]

  1. PMID 34036409, PMID 34357446

Other studies in the field of gynecology and the use of starch mainly concern the prevention of adhesions, which is an undoubted advantage of 4DryField. Kramer et al. showed that the use of 4DryField in endometrial cyst surgery reduces the formation of adhesions by 85% [43]. On the other hand, studies by Ziegler et al. showed that it is safe from the risk of inflammation and is good at preventing the formation of adhesions despite the increase in C-reactive protein levels after surgery [44]

  • The authors should mention that two different modified starch-based agents are available on the market (4DryField and Arista), with 4DryField probably being superior due to its anti-adhesive properties (PMID: 31692813).

It is also important to emphasize that there are currently several starch-based hemostatic products on the market (4DryField, Arista, and others). Not all have been studied in humans, and several involve animal model studies. However, comparing their anti-adhesion effectiveness, 4DryField seems to be the most effective [45]

8) The limitations of the study should be addressed

The limitations of the study were modified.

The limitation of our research is the relatively small group of patients, especially in different subgroups. But the inclusion criteria were demanding, and therefore the recruitment time was relatively long. In our research, we also did not assess the length of time and the number of places that were coagulated, which may be the subject of further analysis. It may also be important to compare different hemostatic substances in terms of their effect on the ovarian reserve. It should be noted that this is a pilot study and should be continued on a larger population

Round 2

Reviewer 3 Report

Thank you for submitting the revised manuscript. In fact, some aspects have been clarified and/or improved. The authors openly admit that patient recruitment was insufficient to obtain reliable patient numbers. In addition, the statement that “obtaining informed consent was not automatic” is hardly comprehensible. Figure 1b does not show the result of the application of the hemostatic powder (= no bleeding before leaving the abdominal cavity), but only the first moment after application with lots of abundant, non-jellified 4DryField. But it's better this than no figure at all. Line 345: “Hemostatic agents” and “tissue sealants” are not the same! Neither TachoSil nor Floseal belong to the category of sealants - please note the classification by Spotnitz & Burks and correct this error before publication. Line 346: Please correct the typo (“Floseal” instead of “Flosil”).

Author Response

We thank you very much for your comments and suggestions regarding our manuscript. Particularly valuable is the reference to the review of hemostatic methods in the publication by Spotnitz & Burks.

The authors openly admit that patient recruitment was insufficient to obtain reliable patient numbers. In addition, the statement that “obtaining informed consent was not automatic” is hardly comprehensible.

The number of patients was small but adequate for the study, however, the time to collect this material was long. This was pointed out by the Reviewer and our explanation is that it was difficult for us to collect the necessary data faster, especially since not all patients with ovarian cysts treated in our clinic agreed to participate in the study. This also explains the remark about the lack of “automatic consent” to participate in the study. This is an explanation to the Reviewer and is not included in the manuscript.

Figure 1b does not show the result of the application of the hemostatic powder (= no bleeding before leaving the abdominal cavity), but only the first moment after application with lots of abundant, non-jellified 4DryField. But it's better this than no figure at all. 

This is in our opinion the best photo after application. The quality of subsequent shots that confirm hemostasis before leaving the abdominal cavity is not so good that would allow them to be published. Thank you for Reviewer accepting our Figure.

Line 345: “Hemostatic agents” and “tissue sealants” are not the same! Neither TachoSil nor Floseal belong to the category of sealants - please note the classification by Spotnitz & Burks and correct this error before publication. 

Line 346: Please correct the typo (“Floseal” instead of “Flosil”).

We have corrected the hemostatic information for TachoSil and FloaSeal. We corrected the spelling of FloaSeal.

There are also reports that compare coagulation with other hemostats or sealants. This mainly applies to TachoSil or FloSeal, i.e., the use of a matrix or gel granules in combination with human thrombin [38]. These studies have shown that all these methods reduce AMH 3 months after surgery, while the decrease is greater with coagulation than with both other methods (41.9% vs. 18.1%, respectively) [38].

Thank you very much for positively accepting our corrections and additions. We hope that the publication of this pilot study will be the beginning of the evaluation of new, needed hemostatic methods in gynecology.